# Characterization of Two Complete Mitochondrial Genomes of *Atkinsoniella* (Hemiptera: Cicadellidae: Cicadellinae) and the Phylogenetic Implications

**DOI:** 10.3390/insects12040338

**Published:** 2021-04-11

**Authors:** Yan Jiang, Hao-Xi Li, Xiao-Fei Yu, Mao-Fa Yang

**Affiliations:** 1Institute of Entomology, Guizhou University, Guiyang 550025, China; yanjianggz@outlook.com; 2Guizhou Provincial Key Laboratory for Agricultural Pest Management of the Mountainous Region, Guiyang 550025, China; lhx_ppath@sina.com (H.-X.L.); anjingfly2009@163.com (X.-F.Y.); 3College of Tobacco Sciences, Guizhou University, Guiyang 550025, China

**Keywords:** leafhopper, mitogenome, *Atkinsoniella*, phylogeny

## Abstract

**Simple Summary:**

*Atkinsoniella* is a large genus of almost 99 species across the world within the subfamily Cicadellinae, which is a large subfamily, comprising more than 2400 species of approximately 330 genera. Some of the Cicadellinae distributed worldwide are known as important agricultural pests. To better understand the mitogenomic characteristics of the genus *Atkinsoniella* and reveal phylogenetic relationships, the complete mitochondrial genomes of *Atkinsoniella grahami* and *Atkinsoniella xanthonota* were sequenced and comparatively analyzed in this study. The mitogenomes of these two *Atkinsoniella* species were found to be highly conserved, similarly to other Cicadellidae, except for the secondary structure of trnaS1, which formed a loop with the dihydrouridine (DHC) arm. This phenomenon has also been observed in other insect mitogenomes. Phylogenetic analyses, based on mitogenomes using both the maximum likelihood (ML) and Bayesian inference (BI) methods of three datasets, supported the monophyly of Cicadellinae, as well as the other subfamilies, and produced a well-resolved framework of Cicadellidae and valuable data for the phylogenetic study of Cicadellinae.

**Abstract:**

The complete mitochondrial genomes of *Atkinsoniella grahami* and *Atkinsoniella xanthonota* were sequenced. The results showed that the mitogenomes of these two species are 15,621 and 15,895 bp in length, with A+T contents of 78.6% and 78.4%, respectively. Both mitogenomes contain 13 protein-coding genes (PCGs), 22 transfer RNA genes (tRNAs), 2 ribosomal RNA genes (rRNAs), and a control region (CR). For all PCGs, a standard start ATN codon (ATT, ATG, or ATA) was found at the initiation site, except for *ATP8*, for which translation is initiated with a TTG codon. All PCGs terminate with a complete TAA or TAG stop codon, except for *COX2*, which terminates with an incomplete stop codon T. All tRNAs have the typical cloverleaf secondary structure, except for *trnS*, which has a reduced dihydrouridine arm. Furthermore, these phylogenetic analyses were reconstructed based on 13 PCGs and two rRNA genes of 73 mitochondrial genome sequences, with both the maximum likelihood (ML) and Bayesian inference (BI) methods. The obtained mitogenome sequences in this study will promote research into the classification, population genetics, and evolution of Cicadellinae insects in the future.

## 1. Introduction

The leafhopper subfamily Cicadellinae is distributed worldwide and contains around 2400 species, represented by approximately 330 genera [1]. In China, 26 genera and 315 species of Cicadellinae have been recorded [2]. Some Cicadellinae insects are of considerable economic importance as they feed on sap in the xylem of woody and herbaceous plants and, via this process, are able to transmit phytopathogenic bacterium and plant viruses to crops, ornamental plants, and weeds [3,4,5,6]. Accurate identification of insects is extremely important for pest control. However, the taxonomic status of some species, based on morphology, remains controversial. Thus, molecular data have been considered as a useful adjunct to the identification and phylogenetic analysis of insects.

The mitochondrial genomes of insects are typically 14.5–17 kb circular double-stranded molecules that contain 37 genes, including 13 protein-coding genes (PCGs), 2 ribosomal RNA genes (rRNAs), 22 transfer RNA genes (tRNAs), and a control region that contains the initial sites for replication and transcription [7,8,9]. With the rapid development of next-generation sequencing technology, the cost of sequencing has been drastically reduced. Mitochondrial genomes are widely used for studying the evolutionary genomics and phylogenetic relationships of different taxonomic levels in insects due to their high genome copy numbers, simple genetic structure [7], maternal inheritance [10], conserved gene components [11], and relatively high evolutionary rate [12]. To date, more than 110 complete or near-complete mitogenomes of Cicadellidae species have been published, most of which have been widely used for evolutionary studies. However, the mitogenomes of only six species of the subfamily Cicadellinae (*Bothrogonia ferruginea*, *B. qiongana*, *Cicadella viridis*, *Cofana yasumatsui*, *Cuerna* sp., and *Homalodisca vitripennis*) (Table 1) have been sequenced and annotated, while none of them belong to the genus *Atkinsoniella*.

*Atkinsoniella* is a large genus within the subfamily Cicadellinae, established with *A*. *decisa* Distant 1908 as its type species, and comprising almost 99 valid species worldwide, with 88 species occurring in China [1,2]. *Atkinsoniella* is mainly distributed in the Oriental realm, with a few in the Palearctic realm. Most of them are polyphagous and often feed on weeds and trees, and they typically live in the damp environment of mountain forests. Their bodies are mostly black with light spots, or light with black, red, yellow, or orange markings. Additionally, they have sexual dimorphism and polymorphism, which makes morphological identification difficult. *A*. *grahami* Young, 1986 (syn. *A*. *nigroscuta* Zhang and Kuoh, 1993 [47] and *A*. *furcula* Yang and Li, 2002 [48]) are widely distributed species in China that have been recorded in several Chinese provinces, including Yunnan, Sichuan, Henan, Shaanxi, Gansu, Hubei, Hunan, Guangdong, Hainan, Chongqing, and Guizhou. Meanwhile, to date, *A. xanthonota* has only been reported in Yunnan province in China [2].

To date, no mitogenome of *Atkinsoniella* has been sequenced. This lack of mitogenomic data has limited the understanding of the evolution of Cicadellinae at the genomic level. Therefore, the mitogenomes of *A. grahami* and *A. xanthonota* were sequenced and analyzed to help us understand the mitogenomic structures and phylogenetic relationships within this group. Hopefully, this study will be valuable for the taxonomy and phylogeny of Cicadellidae insects in the future.

## 2. Materials and Methods

### 2.1. Sample Collection and Mitogenome Sequencing

The specimens of *A*. *grahami* used in this study were collected from the Tangjiahe National Natural Reserve (E: 104°45′48″, N: 32°35′11.8428″, H: 1726 m), Sichuan Province, China on 26 July 2017 (collector: Hong-Li He). The *A*. *xanthonota* specimens were collected from Jinping County (E: 103°13′48″, N: 22°46′55″, H: 1809 m), Yunnan Province, China on 18 July 2018 (collector: Jia-Jia Wang). All of the fresh specimens were immediately preserved in 100% ethanol and stored at −20 °C in the laboratory before DNA extraction. The identification of specimens was based on morphological characteristics, especially the male genitalia, described by Yang et al. [2]. Total genomic DNA was extracted from tissues of the head and thorax muscle of a single adult specimen using a DNeasy^®^ Tissue Kit (Qiagen, Hilden, Germany), according to the manufacturer’s protocol. Voucher DNA (stored at −20 °C) and external genitalia (preserved in glycerol) were deposited at the Institute of Entomology, Guizhou University, Guiyang, China (GUGC). The total genomic DNA of *A. grahami* and *A. xanthonota* were used for library preparation and next-generation sequencing (Illumina NovaSeq6000 platform, Berry Genomic, Beijing, China) with a paired-end 150 sequencing strategy. The clean, next-generation sequencing results were assembled using NOVOPlasty 2.7.2 [49] based on the mitochondrial *COX1* gene fragment (MG397188, submitted by Dewaard) downloaded from GenBank as a starting reference.

### 2.2. Sequence Annotation and Analysis

The initial annotation of the mitogenomes was carried out using Mitoz 2.4-alpha [50], with the invertebrate mitochondrial genetic codes. The locations and secondary structures of the tRNA genes were reconfirmed and predicted using the MITOS web server (http://mitos.bioinf.uni-leipzig.de/index.py, accessed on 6 October 2020) [51] and the tRNAscan-SE search server (http://lowelab.ucsc.edu/tRNAscan-SE/, accessed on 6 October 2020) [52,53]. The start codon, stop codon, and length of the 13 PCGs were manually checked and adjusted by comparison with the published Cicadellinae mitogenome sequences *B. ferruginea* (KU167550) and *H. vitripennis* (NC_006899). Open reading frames (ORFs) were also confirmed based on the invertebrate mitochondrial genetic code. The two rRNA genes were assumed to extend to the boundaries of the locations of adjacent tRNA genes (*trnL1* and *trnV*), and then compared with the homologous rRNA genes of other published Cicadellidae species to define the right boundary of s-rRNA abuts to the control region. The circular mitogenome maps were visualized using OGDRAW (https://chlorobox.mpimp-golm.mpg.de/OGDraw.html, accessed on 9 October 2020) [54]. Strand asymmetry was calculated according to the formulas: AT skew = [A − T] / [A + T] and GC skew = [G − C] / [G + C] [55]. The nucleotide composition statistics and relative synonymous codon usage (RSCU) values of each PCG were computed with MEGA 6.0 [56]. Tandem repeats in the control region were identified using the Tandem Repeats Finder program (http://tandem.bu.edu/trf/trf.html, accessed on 23 October 2020) [57]. The newly sequenced mitogenome sequences of *A. grahami* and *A. xanthonota* were submitted to GenBank with the accession numbers MW533712 and MW533713, respectively.

### 2.3. Phylogenetic Analyses

In the phylogenetic analyses, 65 mitogenomes of 64 available leafhopper species (including two newly sequenced species), representing 11 subfamilies of the family Cicadellidae (Deltocephalinae, Iassinae, Coelidiinae, Macropsinae, Megophthalminae, Idiocerinae, Evacanthinae, Cicadellinae, Typhlocybinae, Ledrinae, and Mileewinae) and six treehopper species of Aetalioninae, Centrotinae, and Smiliinae were selected as the ingroup. In addition, *Callitettix braconoides* (NC_025497) and *Magicicada tredecim* (NC_041652) from the respective families Cercopidae and Cicadidae were used as the outgroup (Table 1). Three datasets were concatenated for phylogenetic analysis: (1) AA: amino acid sequences of the PCGs; (2) PCG12: first and second codon positions of the PCGs; (3) PCG12RNA: the first and the second codon positions of the PCGs and two rRNA genes. Since some rRNA genes of the mitogenome sequences were unavailable, the number of species used for the PCG12RNA dataset was different to that of the PCG12 and AA datasets.

The alignments of 13 PCGs (without stop codons) were aligned using the MASCE [58] algorithm in PhyloSuite 1.2.1 [59], with the invertebrate mitochondrial genetic code. The rRNA genes were aligned with MAFFT 7.313 [60] using the Q-INS-I strategy; gaps and ambiguous sites were removed using the Gblocks 0.91b [61,62] algorithm in PhyloSuite 1.2.1 and default settings, with the exception of AA gap positions, which were toggled to “none”. Then, the alignments of each individual gene were concatenated as different datasets using Geneious prime 2020.2.4.

The best schemes for the partition and substitution models were determined in PartitionFinder v.2.1.1 with the corrected Akaike information criterion (AICc) and greedy search algorithm [63]. The starting partitions used to initiate the PartitionFinder analysis are listed in Appendix A. Maximum likelihood (ML) analysis was conducted using IQ-TREE v.1.6.8 [64]. Branch support was estimated with 10,000 replicates of ultrafast bootstrap. Bayesian inference (BI) analysis was performed on MrBayes 3: Bayesian phylogenetic inference under mixed [65] with the default settings, by simulating four independent runs for 100 million generations with sampling every 1000 generations, the initial 25% of samples were discarded as burn-in.

## 3. Results and Discussion

### 3.1. Mitogenome Organization and Nucleotide Composition

Nine complete or partial mitogenomes were analyzed: the two mitogenomes newly sequenced in this study and another seven mitogenomes downloaded from GenBank without any revision of annotations. The length of the entire mitogenome sequences ranged from 12,696 to 15,895 bp, and contained 37 genes (13 PCGs, 22 tRNA genes, and 2 rRNA genes) and a control region. The *Cuerna* sp. (KX437741) and *Cicadella viridis* (KY752061) mitogenomes were incomplete. The *A. grahami* and *A. xanthonota* mitogenomes are closed circular molecules 15,621 and 15,895 bp in length, respectively (Figure 1 and Figure 2). The variation in mitogenome size among the different Cicadellinae insects is mainly due to the variable number of repeats in the control region. These nine Cicadellinae mitogenome sequences showed identical gene orders, with the typical gene arrangement of insects. A total of 14 genes: four PCGs (*ND5*, *ND4*, *ND4L*, and *ND1*), eight tRNAs (*trnQ*, *trnC*, *trnY*, *trnF*, *trnH*, *trnT*, *trnP*, and *trnL*), and two rRNAs (*l-rRNA* and *s–rRNA*), were encoded on the minority strand (N-strand), while the other 23 genes (nine PCGs and 14 tRNAs) were encoded on the majority strand (J-strand) (Table 2 and Table 3).

The overall nucleotide composition of *A. grahami* was determined as A: 41.9%, T: 36.6%, C: 11.6%, and G: 9.9%, while it was A: 41.8%, T: 36.7%, C: 11.7%, and G: 9.9% in *A. xanthonota*. Similar to the other Cicadellinae mitogenomes, these two mitogenomes were both consistently AT nucleotide biased, with 78.6% in *A. grahami* and 78.4% in *A. xanthonota*. The A+T content of the rRNAs was the highest (82.1% in *A. grahami* and 81.9% in *A. xanthonota*), while the A+T content of the PCGs was the lowest (77.5% in *A. grahami* and 77.4% in *A. xanthonota*) (Table 4). The AT skew was 0.068 for *A.*
*grahami* and 0.065 for *A. xanthonota*, indicating a slightly higher occurrence of A compared to T nucleotides. Similar results were also observed for the entire mitogenomes of the seven other Cicadellinae, where all of the AT skews were positive and all of the GC skews were negative (Table 5).

### 3.2. Overlapping and Intergenic Spacer Regions

The mitogenomes of *A.*
*grahami* and *A. xanthonota* were found to have a total of 66 bp (sum of 15 locations) and 64 bp (sum of 14 locations) overlaps between genes, respectively. The longest observed overlaps were both 20 bp, located between the *trnF* and *ND5* genes. The total length of the intergenic spacers was 38 bp for *A.*
*grahami* and 41 bp for *A. xanthonota*. All of the intergenic spacers of the two mitogenomes range from 1 to 15 bp. The longest intergenic spacer situated between *trnS* and *ND1* was found in both the *A.*
*grahami* and *A. xanthonota* mitogenomes. There is a 2 bp overlap in *A.*
*grahami* and a 3 bp intergenic spacer region in *A. xanthonota* between *CYTB* and *trnS*. Aside from this difference, all of the overlapping and intergenic spacers were identical.

### 3.3. Protein-Coding Genes and Codon Usage

The 13 PCGs of *A. grahami* encode 3645 amino acids with a total length of 10,935 bp (excluding the stop codons), and those of *A. xanthonota* encode 3643 amino acids with a total length of 10,929 bp (excluding the stop codons). The A+T contents of the 13 PCGs were 77.5% and 77.4% in the in *A. grahami* and *A. xanthonota* mitogenomes, respectively. Moreover, the A+T content of the third codon position was higher than that of the first and second codon positions in these newly sequenced mitogenomes. The 13 PCGs in *A. grahami* and *A. xanthonota* both showed a negative AT skew (−0.150 and −0.152, respectively) and a positive GC skew (0.012 and 0.013, respectively). Nine PCGs were encoded on the J-strand and four PCGs (*ND5*, *ND4*, *ND4L*, and *ND1*) were encoded on the N-strand.

These two newly sequenced *Atkinsoniella* mitogenomes exhibited similar start and stop codons. Translation of all PCGs is initiated with typical ATN codons (ATT, ATG, or ATA), except for *ATP8*, which is initiated with a TTG codon (Table 2 and Table 3). While 12 PCGs terminated with the complete stop codon TAA/TAG, the truncated stop codon T was also detected in the *COX2* gene in the two *Atkinsoniella* mitogenomes. Such an incomplete stop codon is commonly found in insect mitogenomes and is presumed to be generated by the polyadenylation process [9]. Meanwhile, the stop codons of *ATP8* and *CYTB* differed between these two species. *A. grahami* used TAA and TAG as the stop codons for *ATP8* and *CYTB*, respectively, while *A. xanthonota* used TAG and TAA as the stop codons of *ATP8* and *CYTB*, respectively. In the seven other sequenced Cicadellinae mitogenomes that were examined, similar start and stop codons were found, except for the incomplete stop codon observed in *COX1* of *Homalodisca vitripennis*, *ND3* and *CYTB* of *B. ferruginea*, and *COX3* of *Cicadella viridis* (Appendix A). In these nine mitogenomes, the termination codon TAA occurred more frequently than TAG. Meanwhile, all *COX2* ended with incomplete T or TA. The relative synonymous codon usage (RSCU) of nine sequenced Cicadellinae mitogenomes (of eight species) was calculated and drawn (Figure 3). The results show that the nine mitogenomes share the same codon families and similar RSCU features. The four most frequently used codons observed for these nine mitogenomes are UUU-Phe, UUA-Leu, AUU-Ile, and AUA-Met, which were composed of A and U.

### 3.4. Transfer and Ribosomal RNA Genes

The two *Atkinsoniella* mitogenomes both contain the typical number of 22 tRNAs, eight of which were encoded by N-strand and 14 encoded by J-strand, ranging from 61 bp (trnA, trnH) to 71 bp (trnK) in length (Table 2 and Table 3). The total length of tRNA sequence was 1426 and 1423 bp, accounting for 9.13% and 8.95% of the whole mitogenome in *A. grahami* and *A. xanthonota*, respectively. All tRNAs of these two mitogenomes indicated a positive AT skew (0.022 and 0.018, respectively) and a positive GC skew (0.152 and 0.137, respectively). Most tRNAs exhibited typical cloverleaf secondary structures, except for trnS1 (GCU), which lacks a dihydrouridine (DHU) arm, and instead being replaced by a simple loop. Moreover, there were some kinds of unmatched base pairs (Figure 4 and Figure 5). These two structural characteristics are also present in other leafhoppers [40,48,49,66,67,68,69]. In the predicted secondary structure, the length of the anticodon loop of all tRNAs was highly conserved for 7 bp, compared to the variable sizes of the DHU and TΨC loops.

Two rRNA genes (*l-rRNA* and *s-rRNA*) were recognized in the two newly sequenced mitogenomes. *l-rRNA* was located between the *trnL1* and *trnV*. The *s-rRNA* was located between *trnV* and the control region. The length of two rRNA genes were 1215 bp (*l-rRNA*) and 740 bp (*s-rRNA*) in *A. grahami,* and 1217 bp *(l-rRNA*) and 798 bp (*s-rRNA*) in *A. xanthonota*. The A+T content region of the rRNAs in Cicadellinae mitogenomes ranged from 78.3% to 82.1% (Table 5). The A+T content reached 82.1% in *A. grahami* and 81.9% in *A. xanthonota*. Additionally, the two rRNAs in these two mitogenomes showed a negative AT skew (−0.119 in *A. grahami* and −0.117 in *A. xanthonota*) and a positive GC skew (0.234 in *A. grahami* and 0.231 in *A. xanthonota*).

### 3.5. Control Region

The control region is the longest non-coding region, plays an indispensable role in the study of molecular evolution, and contains regulatory functions for replication and transcription [7,8,70]. The control regions in the Cicadellinae mitogenomes were not highly conserved and were located between the *s-rRNA* and *trnI* genes, and ranged from 658 to 1645 bp in size (Figure 6). The length of the control regions in the two *Atkinsoniella* mitogenomes was 1296 bp in *A. grahami* and 1514 bp in *A. xanthonota* (Figure 1, Table 2 and Table 3). The A+T content was 80.4% and 79.9%, respectively. The AT and GC skews were both positive, with values of 0.052 and 0.071 in *A. grahami,* and 0.029 and 0.118 in *A. xanthonota*, indicating that A and G were more abundant than T and C. The Cicadellinae mitogenomes had one to three types of tandem repeat units, ranging from 14 to 220 bp. One tandem repeat (212 bp) was found in the control region of the *A. grahami* mitogenome, and two tandem repeats (220 and 14 bp) were found in the *A. xanthonota* mitogenome (Figure 6).

Furthermore, poly-A regions were found at the end of the control region in *A. grahami*, *A. xanthonota,* and *B. ferruginea*. This has also been observed in other insect mitogenomes [71,72,73,74,75]. Two and five poly-T regions were also found in *A. grahami* and *A. xanthonota*, respectively. The results indicate that the length, nucleotide sequences, and copy numbers of the tandem repeat units in the control region were variable among known Cicadellinae mitogenomes.

### 3.6. Phylogenetic Analyses

ML and BI analyses were used to reconstruct the phylogenetic relationships among the 65 species of the 11 subfamilies of leafhoppers, 6 species of treehoppers, and 2 outgroups, under the best partitioning scheme and models selected by PartitionFinder (Appendix A). Six phylogenetic trees (ML-AA, BI-AA, ML-PCG12, BI-PCG12, ML-PCG12RNA, and BI-PCG12RNA) were established, with most nodes receiving high nodal support values and a few nodes receiving only moderate or low support in the ML and BI trees (Figure 7 and Figure 8 and Appendix A). In the obtained topology, each subfamily was consistently recovered as monophyletic in different analyses, while the relationships among subfamilies were discrepant across analyses. The comparative analysis of these six phylogenetic trees showed that treehoppers share a common ancestor with the subfamilies Deltocephalinae, Iassinae, Coelidiinae, Macropsinae, Megophthalminae, Idiocerinae, Evacanthinae, Cicadellinae, Typhlocybinae, Ledrinae, and Mileewinae. The results of this study support the point that treehoppers are derived from paraphyletic Cicadellidae, which has been reported in former studies [30,37,38,76,77,78].

Moreover, within the family Cicadellidae, some relationships were highly supported and constant. Iassinae and Coelidiinae were sister groups with high support values (bootstrap support values (BS) ≥ 96, Bayesian posterior probability (PP) = 1) among the six phylogenetic trees, which is consistent with the results of previous studies based on mitogenomes [26,31,32,33]. And Macropsinae emerged as a sister group with Iassinae and Coelidiinae in all phylogenetic trees except for the BI-AA tree. Meanwhile, most nodes within each subfamily received high support and the relationships within Coelidiinae, Typhlocybinae, and Mileewinae were consistent in all analyses. 

Within the subfamily Cicadellinae, the phylogenetic relationships indicated that Cicadellinae was consistently a monophyletic group. This is different from the previous studies based on mitogenomes reported by Wang [30,33], which suggested Cicadellinae was not a monophyletic group. All of the ML and BI analyses suggested that the relationships within Cicadellinae were ((*H. vitripennis* + (*Co. yasumatsui* + *Ci. viridis*)) + ((*B. ferruginea* + *B. qiongana*) + (*Cuerna* sp. + (*A. grahami* + *A*. *xanthonota*)))) based on AA and PCG12, and ((*H. vitripennis* + (*Co. yasumatsui* + *Ci. viridis*)) + ((*B. ferruginea* + *B. qiongana*) + (*A. grahami* + *A*. *xanthonota*))) based on PCG12RNA with high support values (Figure 8 and Appendix A). The newly sequenced *A. grahami* grouped with *A*. *xanthonota*. The inferred relationships among the genus of Cicadellinae were highly stable; *Homalodisca*, *Cofana,* and *Cicadella* clustered to a branch, forming a sister group with the clade that *Atkinsoniella* and *Bothrogonia* formed. Additionally, Mileewinae and Cicadellinae were divided into two separate clades in this study within all phylogenetic trees, except for the BI-PCG12 tree, which formed polytomies, potentially due to inadequate data for ascertaining how these lineages are related (Figure 7and Appendix A). These results support the point of Young [79], and Linnavuori and Delong [80] that Mileewinae is a separate group from Cicadellinae. 

## 4. Conclusions

In this study, the complete mitochondrial genome sequences of *A*. *grahami* and *A*. *xanthonota* were provided, with a comparative analysis within the available mitogenome sequences of Cicadellinae and a phylogenetic analysis of Cicadellidae. This is the first report of mitogenome sequences from the genus *Atkinsoniella* of subfamily Cicadellinae. The lengths of these two mitogenomes were 15,621 and 15,895 bp, for *A*. *grahami* and *A*. *xanthonota*, respectively. Comparison with other previously reported mitogenomes of Cicadellinae showed that all had a similar structural characteristics and nucleotide compositions. All of the Cicadellinae mitogenomes were highly conserved in holistic organization, exhibiting the same gene order as hypothetical ancestral insects, and all mitogenomes were composed of 37 typically encoded genes and a control region, except for *Cuerna* sp. (KX437741) and *Cicadella viridis* (KY752061), which lacked *trnV*, *s-rRNA*, and a control region, due to the incomplete mitogenomes sequences. These two newly sequenced mitogenomes of genus *Atkinsoniella* can provide valuable data for future studies of phylogenetic relationships of Cicadellinae.

Maximum likelihood and Bayesian inference analyses among the major lineages based on the concatenated alignments of AA, PCG12, and PCG12RNA indicated that each subfamily of leafhoppers (Cicadellinae) and treehoppers (Membracidae) was recovered as monophyletic group, and that the treehoppers originated from paraphyletic Cicadellidae, as per previous reports. The analyses produced a well-resolved framework for the relationships within each subfamily. The relationships within subfamily Cicadellinae, Typhlocybinae, Mileewinae, and Coelidiinae were stable in all phylogenetic trees with high support. However, a few deep nodes received low or moderate support values and the phylogenetic relationship among subfamilies was not well resolved, which may have been restricted by the limited sampling molecular data in this study. Perhaps, more mitogenomic taxon sampling, richer molecular data, and a combined approach of mitogenomes and nuclear markers would elucidate the unresolved relationships among these subfamilies and help to understand the phylogenetic and evolutionary relationships within Cicadellidae.

## Figures and Tables

**Figure 1 insects-12-00338-f001:**
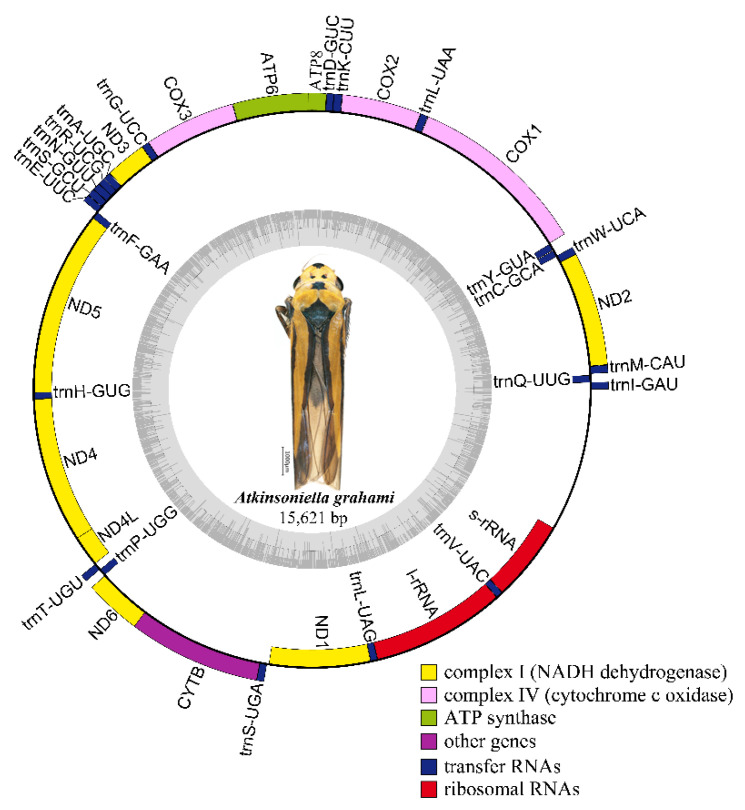
Circular map of the mitochondrial genome of *Atkinsoniella grahami.* Genes are represented by different color blocks. Color blocks outside the circle indicate that the genes are on the majority strand (J-strand); those within the circle indicate the genes are located on the minority strand (N-strand).

**Figure 2 insects-12-00338-f002:**
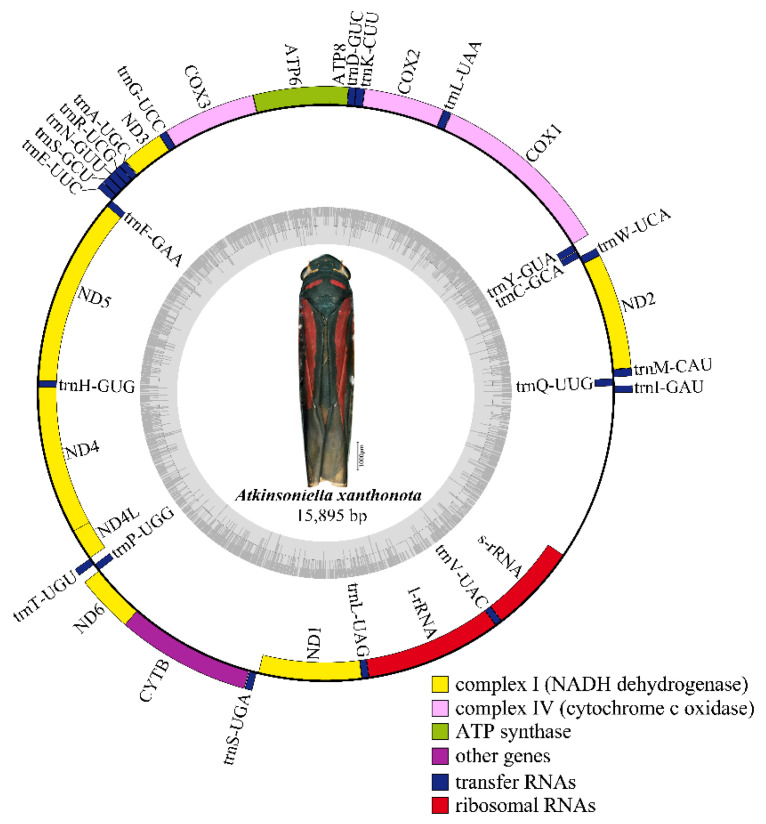
Circular map of the mitochondrial genome of *Atkinsoniella xanthonota.* Genes are represented with different color blocks. Color blocks outside the circle indicate the genes are on the majority strand (J-strand); those within the circle indicate the genes are located on the minority strand (N-strand).

**Figure 3 insects-12-00338-f003:**
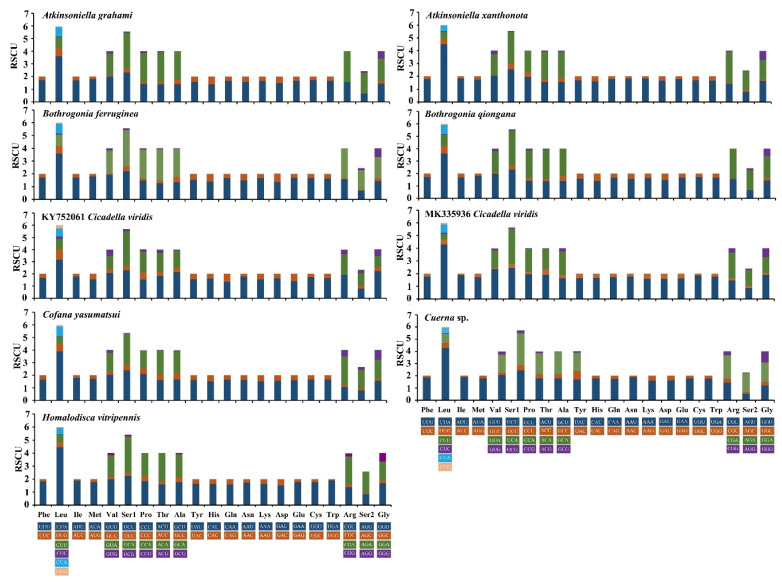
Relative synonymous codon usage (RSCU) in protein-coding genes (PCGs) of the mitogenomes of *Atkinsoniella grahami*, *Atkinsoniella xanthonota,* and other six Cicadellinae species. Codon families are indicated in boxes below the *x*-axis; the colors correspond to the stacked columns, and frequency of RSCU is provided on the *y*-axis.

**Figure 4 insects-12-00338-f004:**
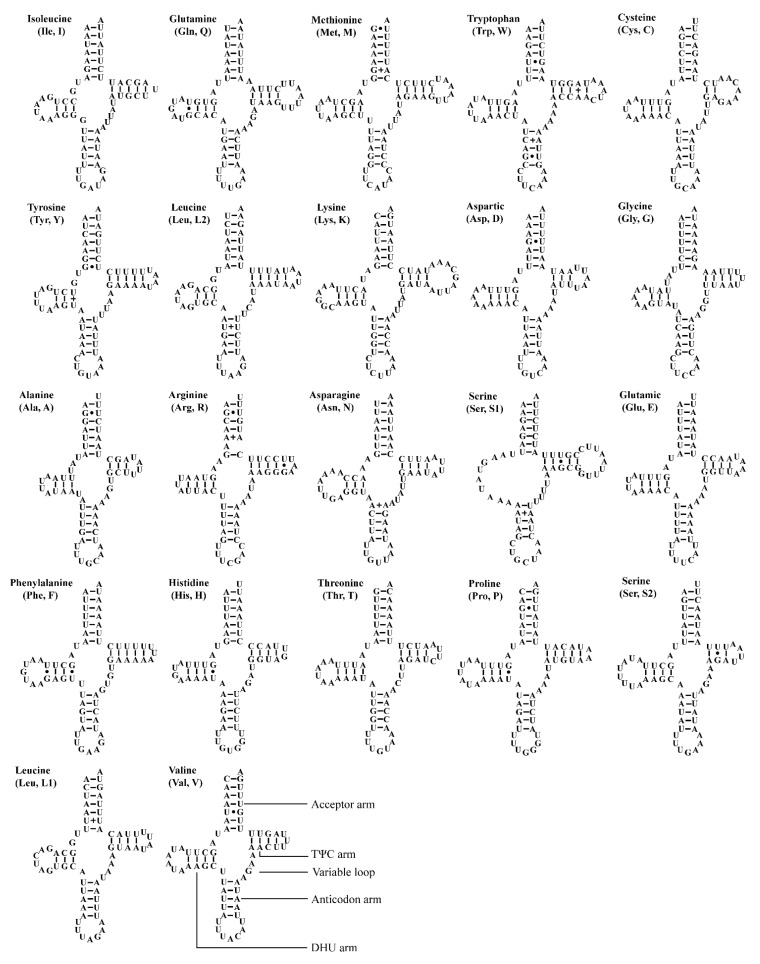
Predicted secondary cloverleaf structure for the tRNAs of *Atkinsoniella grahami*. The tRNA arms are illustrated as for trnV. Dashes (–), solid dots (•), and pluses (+) indicate the Watson–Crick base pairings, G–U bonds, and mismatches, respectively.

**Figure 5 insects-12-00338-f005:**
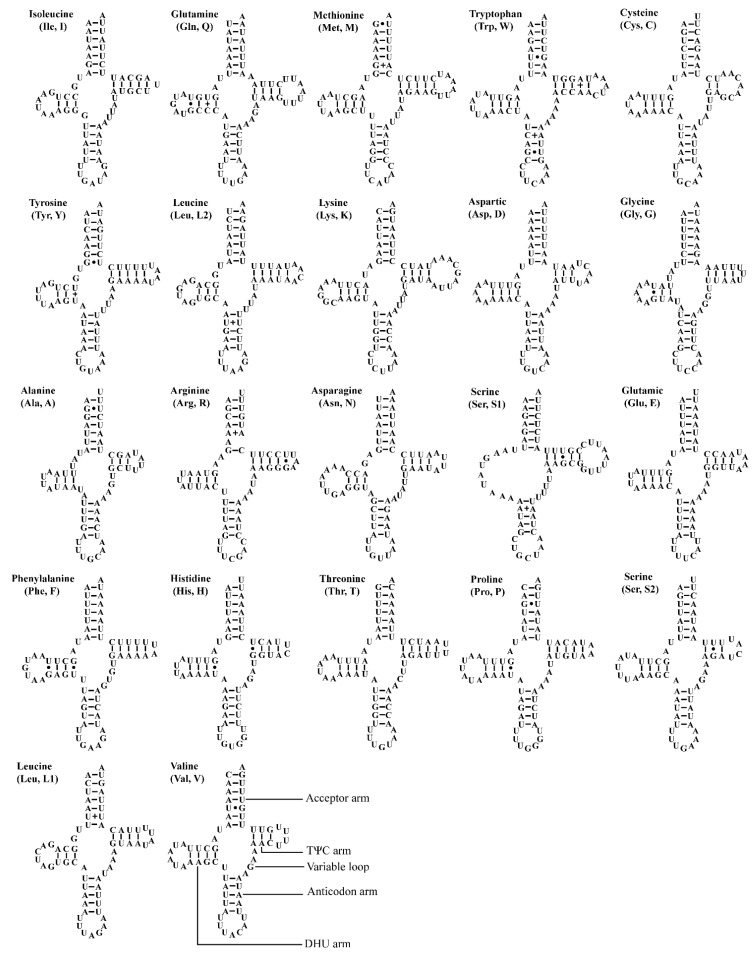
Predicted secondary cloverleaf structure for the tRNAs of *Atkinsoniella xanthonota*. The tRNA arms are illustrated as for trnV. Dashes (–), solid dots (•), and pluses (+) indicate the Watson–Crick base pairings, G–U bonds, and mismatches, respectively.

**Figure 6 insects-12-00338-f006:**
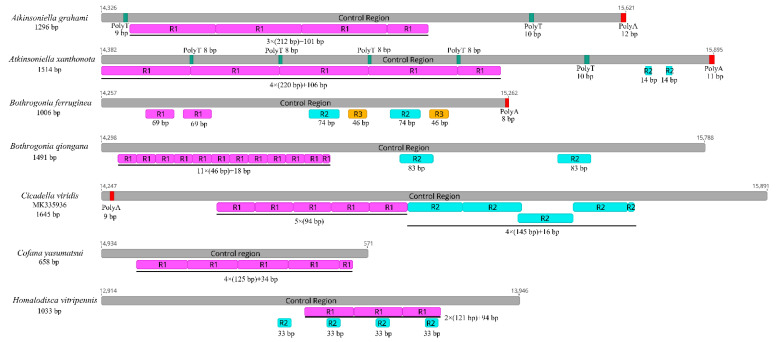
Organization of the control regions in the mitochondrial genomes of *Atkinsoniella graham*, *Atkinsoniella xanthonota,* and the five other Cicadellinae species. The gray blocks refer to the control region. The numbers located at the start and end of gray blocks refer to the location of control region in each mitogenome. R refers to repeat unit, with the number indicating the number of repeats. The red and green blocks refer to the structures of poly (A) and poly (T), respectively.

**Figure 7 insects-12-00338-f007:**
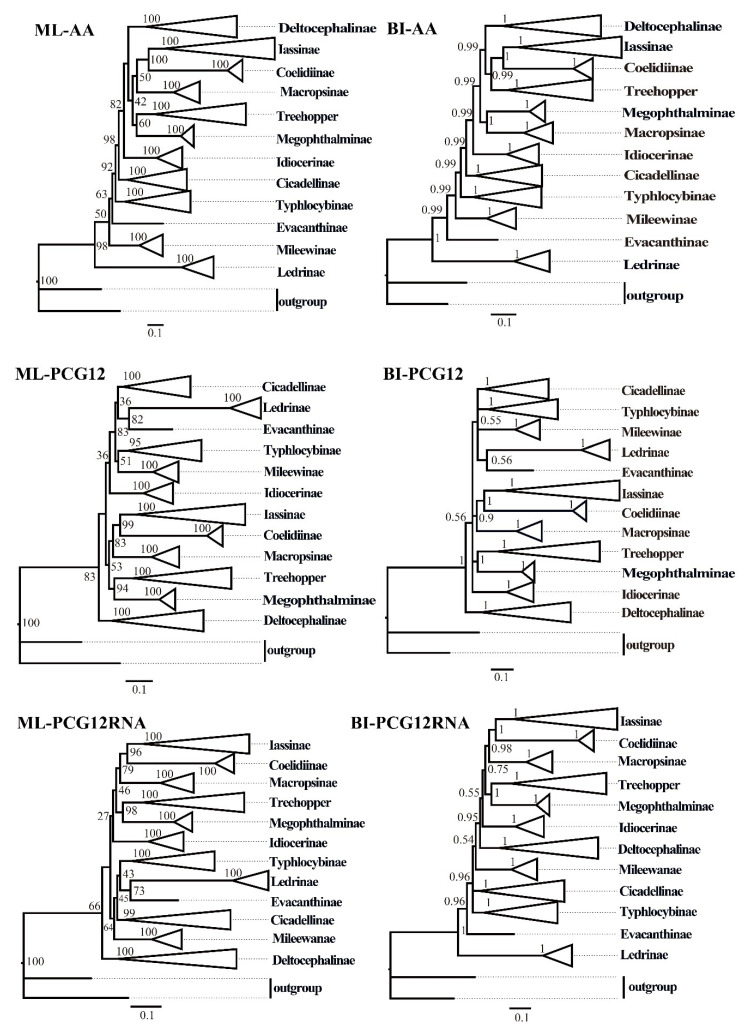
The phylogenetic trees of leafhoppers were inferred from different mitochondrial genome datasets using maximum likelihood (ML) and Bayesian inference (BI) methods. AA: amino acid sequences of the protein-coding genes (PCGs) from 73 species; PCG12: first and second codon positions of PCGs from 73 species; PCG12RNA: the first and the second codon positions of the PCGs and two rRNA genes from 67 species. Numbers on each node correspond to the bootstrap support values (BS) and Bayesian posterior probability (PP).

**Figure 8 insects-12-00338-f008:**
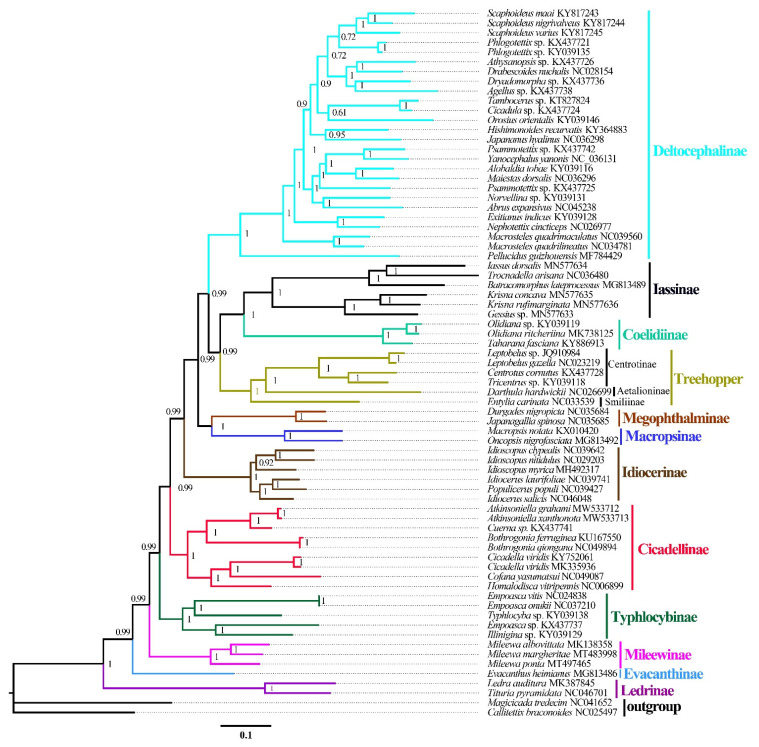
Phylogenetic trees inferred by Bayesian inference (BI) based on the amino acids (AA). Numbers on each node correspond to the Bayesian posterior probability (PP) for 100 million generations.

**Table 1 insects-12-00338-t001:** Mitochondrial genomes used for the phylogenetic analyses in this study.

Subfamily	Species	Size	A+T (%)	Accession Number	Reference
Cicadellinae	*Bothrogonia ferruginea*	15,262	76.5	KU167550	[13]
	*Bothrogonia qiongana*	15,304	78.4	NC_049894	[14]
	*Cuerna* sp.	12,696	77.5	KX437741	[15]
	*Cicadella viridis*	13,461	78.8	KY752061	Unpublished
	*Cicadella viridis*	15,891	78.1	MK335936	[16]
	*Homalodisca vitripennis*	15,304	78.4	NC_006899	Unpublished
	*Cofana yasumatsui*	15,019	80.6	NC_049087	[17]
	*Atkinsoniella grahami*	15,621	78.6	MW533713	This Study
	*Atkinsoniella xanthonota*	15,895	78.3	MW533712	This Study
Mileewinae	*Mileewa albovittata*	15,079	79.6	MK138358	[18]
	*Mileewa margheritae*	15,375	79.2	MT483998	[19]
	*Mileewa ponta*	15,999	79.9	MT497465	[20]
Ledrinae	*Tituria pyramidata*	15,331	75.6	NC_046701	[21]
	*Ledra auditura*	16,094	76.3	MK387845	[22]
Typhlocybinae	*Illinigina* sp.	14,803	76	KY039129	[23]
	*Typhlocyba* sp.	15,223	77.1	KY039138	[23]
	*Empoasca onukii*	15,167	78.3	NC_037210	Unpublished
	*Empoasca vitis*	15,154	78.3	NC_024838	[24]
	*Empoasca* sp.	15,116	76.8	KX437737	[15]
Evacanthinae	*Evacanthus heimianus*	15,806	79.9	MG813486	[25]
Idiocerinae	*Populicerus populi*	16,494	77.2	NC_039427	[26]
	*Idiocerus salicis*	16,436	77.3	NC_046048	[26]
	*Idiocerus laurifoliae*	16,811	79.5	NC_039741	[26]
	*Idioscopus myrica*	15,423	77.9	MH492317	[26]
	*Idioscopus clypealis*	15,393	78.3	NC_039642	[27]
	*Idioscopus nitidulus*	15,287	78.6	NC_029203	[28]
Megophthalminae	*Durgades nigropicta*	15,974	78.8	NC_035684	[29]
	*Japanagallia spinosa*	15,655	76.6	NC_035685	[29]
Macropsinae	*Oncopsis nigrofasciata*	15,927	79	MG813492	[30]
	*Macropsis notata*	16,323	76.8	NC_042723	[30]
Coelidiinae	*Olidiana* sp.	15,253	78.1	KY039119	Unpublished
	*Taharana fasciana*	15,161	77.9	NC_036015	[31]
	*Olidiana ritcheriina*	15,166	78	NC_045207	[32]
Iassinae	*Batracomorphus lateprocessus*	15,356	80.4	MG813489	[33]
	*Trocnadella arisana*	15,131	80.7	NC_036480	[33]
	*Iassus dorsalis*	15,176	80.1	MN577634	[33]
	*Krisna concava*	14,304	79.8	MN577635	[33]
	*Krisna rufimarginata*	14,724	81.1	MN577636	[33]
	*Gessius rufidorsus*	14,634	80.7	MN577633	[33]
Deltocephalinae	*Maiestas dorsalis*	15,352	78.7	NC_036296	[34]
	*Japananus hyalinus*	15,364	76.6	NC_036298	[34]
	*Alobaldia tobae*	16,026	77.3	KY039116	[23]
	*Psammotettix* sp.	12,970	74.7	KX437742	[15]
	*Psammotettix* sp.	12,913	76.7	KX437725	[15]
	*Yanocephalus yanonis*	15,623	74.6	NC_036131	[23]
	*Abrus expansivus*	15,904	74.7	NC_045238	[35]
	*Norvellina* sp.	15,594	74.5	KY039131	[23]
	*Nephotettix cincticeps*	14,805	77.6	NC_026977	Unpublished
	*Exitianus indicus*	16,089	75.1	KY039128	[23]
	*Macrosteles quadrilineatus*	16,626	78	NC_034781	[36]
	*Macrosteles quadrimaculatus*	15,734	77.7	NC_039560	[37]
	*Hishimonoides recurvatis*	14,814	76.7	KY364883	Unpublished
	*Scaphoideus maai*	15,188	77.2	KY817243	[38]
	*Scaphoideus nigrivalveus*	15,235	76.6	KY817244	[38]
	*Scaphoideus varius*	15,207	75.9	KY817245	[38]
	*Phlogotettix* sp.	15,136	77.9	KY039135	[23]
	*Phlogotettix* sp.	12,794	77	KX437721	[15]
	*Drabescoides nuchalis*	15,309	75.6	NC_028154	[39]
	*Agellus* sp.	14,819	75.8	KX437738	[15]
	*Athysanopsis* sp.	14,573	74.1	KX437726	[15]
	*Dryadomorpha* sp.	12,297	74.1	KX437736	[15]
	*Tambocerus* sp.	15,955	76.4	KT827824	[40]
	*Cicadula* sp.	14,929	74.1	KX437724	[15]
	*Orosius orientalis*	15,513	72	KY039146	[23]
	*Pellucidus guizhouensis*	16,555	78	MF784429	Unpublished
Centrotinae	*Centrotus cornutus*	14,696	76.9	KX437728	[15]
	*Tricentrus* sp.	15,419	78.5	KY039118	Unpublished
	*Leptobelus gazella*	16,007	78.8	NC_023219	[41]
	*Leptobelus* sp.	15,201	77.5	JQ910984	[42]
Smiliinae	*Entylia carinata*	15,662	78.1	NC_033539	[43]
Aetalioninae	*Darthula hardwickii*	15,355	78	NC_026699	[44]
outgroup	*Callitettix braconoides*	15,637	77.2	NC_025497	[45]
	*Magicicada tredecim*	14,435	76.3	NC_041652	[46]

**Table 2 insects-12-00338-t002:** Annotations for the *Atkinsoniella grahami* mitochondrial genomes.

Gene	Direction	Location	Anticodon	Size (bp)	Start Codon	Stop Codon	Intergenic Nucleotides
*trnI*	J	1–63	GAU	63			
*trnQ*	N	61–128	UUG	68			−3
*trnM*	J	139–206	CAU	68			10
*ND2*	J	207–1178		972	ATT	TAA	0
*trnW*	J	1177–1244	UCA	68			−2
*trnC*	N	1237–1299	GCA	63			−8
*trnY*	N	1303–1367	GUA	65			3
*COX1*	J	1371–2906		1536	ATG	TAA	3
*trnL*	J	2908–2972	UAA	65			1
*COX2*	J	2973–3651		679	ATT	T	0
*trnK*	J	3652–3722	CUU	71			0
*trnD*	J	3722–3784	GUC	63			−1
*ATP8*	J	3785–3937		153	TTG	TAA	0
*ATP6*	J	3931–4581		651	ATG	TAA	−7
*COX3*	J	4582–5361		780	ATG	TAA	0
*trnG*	J	5361–5423	UCC	63			−1
*ND3*	J	5424–5777		354	ATT	TAA	0
*trnA*	J	5780–5840	UGC	61			2
*trnR*	J	5841–5902	UCG	62			0
*trnN*	J	5902–5967	GUU	66			−1
*trnS*	J	5967–6032	GCU	66			−1
*trnE*	J	6033–6095	UUC	63			0
*trnF*	N	6095–6161	GAA	67			−1
*ND5*	N	6142–7839		1698	ATT	TAA	−20
*trnH*	N	7837–7897	GUG	61			−3
*ND4*	N	7897–9219		1320	ATG	TAA	−1
*ND4L*	N	9213–9494		282	ATG	TAA	−7
*trnT*	J	9497–9561	UGU	65			2
*trnP*	N	9562–9627	UGG	66			0
*ND6*	J	9630–10,118		489	ATT	TAA	2
*CYTB*	J	10,111–11,247		1137	ATG	TAG	−8
*trnS*	J	11,246–11,310	UGA	65			−2
*ND1*	N	11,326–12,243		918	ATT	TAA	15
*trnL*	N	12,244–12,306	UAG	63			0
*l–rRNA*	N	12,307–13,521		1215			0
*trnV*	N	13,522–13,585	UAC	64			0
*s–rRNA*	N	13,586–14,325		740			0
Control region		14,326–15,621		1296			

**Table 3 insects-12-00338-t003:** Annotations for the *Atkinsoniella xanthonota* mitochondrial genomes.

Gene	Direction	Location	Anticodon	Size (bp)	Start Codon	Stop Codon	Intergenic Nucleotides
*trnI*	J	1–63	GAU	63			
*trnQ*	N	61–128	UUG	68			−3
*trnM*	J	139–206	CAU	68			10
*ND2*	J	207–1178		972	ATT	TAA	0
*trnW*	J	1177–1244	UCA	68			−2
*trnC*	N	1299–1237	GCA	63			−8
*trnY*	N	1367–1303	GUA	65			3
*COX1*	J	1371–2906		1536	ATG	TAA	3
*trnL*	J	2908–2972	UAA	65			1
*COX2*	J	2973–3651		679	ATT	T	0
*trnK*	J	3652–3722	CUU	71			0
*trnD*	J	3722–3784	GUC	63			−1
*ATP8*	J	3785–3937		153	TTG	TAG	0
*ATP6*	J	3931–4581		651	ATG	TAA	−7
*COX3*	J	4582–5361		780	ATG	TAA	0
*trnG*	J	5361–5423	UCC	63			−1
*ND3*	J	5424–5777		354	ATT	TAA	0
*trnA*	J	5780–5840	UGC	61			2
*trnR*	J	5841–5902	UCG	62			0
*trnN*	J	5902–5967	GUU	66			−1
*trnS*	J	5967–6032	GCU	66			−1
*trnE*	J	6033–6095	UUC	63			0
*trnF*	N	6160–6095	GAA	66			−1
*ND5*	N	7838–6141		1698	ATT	TAA	−20
*trnH*	N	7896–7836	GUG	61			−3
*ND4*	N	9218–7896		1323	ATG	TAA	−1
*ND4L*	N	9493–9212		282	ATG	TAA	−7
*trnT*	J	9496–9560	UGU	65			2
*trnP*	N	9626–9561	UGG	66			0
*ND6*	J	9629–10,117		489	ATT	TAA	2
*CYTB*	J	10,110–11,240		1131	ATG	TAA	−8
*trnS*	J	11,244–11,307	UGA	64			3
*ND1*	N	12,240–11,323		918	ATT	TAA	15
*trnL*	N	12,303–12,241	UAG	63			0
*l–rRNA*	N	13,520–12,304		1217			0
*trnV*	N	13,583–13,521	UAC	63			0
*s–rRNA*	N	14,381–13,584		798			0
Control region		14,382–15,895		1514			

**Table 4 insects-12-00338-t004:** Nucleotide composition of the *Atkinsoniella grahami* and *Atkinsoniella xanthonota* mitogenomes.

Species	Regions	Length (bp)	T%	C%	A%	G%	A+T%	AT Skew	GC Skew
*A. grahami*	Whole genome	15,621	36.6	11.6	41.9	9.9	78.6	0.068	−0.080
	PCGs *	10,972	44.6	11.1	32.9	11.4	77.5	−0.150	0.012
	1st codon position **	3657	38.3	10.6	35.4	15.8	73.6	−0.040	0.199
	2nd codon position **	3657	48.1	17.1	20.9	13.9	69.0	−0.394	−0.101
	3rd codon position **	3657	47.4	5.7	42.5	4.3	90.0	−0.054	−0.131
	tRNAs ***	1426	39.2	8.4	41.0	11.4	80.2	0.022	0.152
	rRNAs ****	1955	45.9	6.9	36.2	11.0	82.1	−0.119	0.234
	Control region	1296	38.1	9.1	42.3	10.5	80.4	0.052	0.071
*A. xanthonota*	Whole genome	15,894	36.7	11.7	41.8	9.9	78.4	0.065	−0.082
	PCGs *	10,966	44.6	11.2	32.8	11.5	77.4	−0.152	0.013
	1st codon position **	3655	38.3	10.5	35.4	15.8	73.8	−0.039	0.203
	2nd codon position **	3655	48.0	17.0	21.0	13.9	69.1	−0.391	−0.103
	3rd codon position **	3655	47.3	6.0	42.0	4.7	89.3	−0.059	−0.118
	tRNAs ***	1423	39.3	8.6	40.7	11.4	80.0	0.018	0.137
	rRNAs ****	2015	45.8	6.9	36.2	11.1	81.9	−0.117	0.231
	Control region	1513	38.8	8.9	41.1	11.2	79.9	0.029	0.118

* PCGs: protein-coding genes; ** 1st, 2nd, 3rd codon position: the 1st, 2nd, 3rd codon position of the PCGs; *** tRNAs: transfer RNA genes; **** rRNAs: ribosomal RNA genes.

**Table 5 insects-12-00338-t005:** Nucleotide composition of the Cicadellinae mitochondrial genomes of *Atkinsoniella grahami*, *Atkinsoniella xanthonota*, *Bothrogonia ferruginea*, *Bothrogonia qiongana*, *Cicadella viridis*, *Cofana yasumatsui*, *Cuerna* sp., and *Homalodisca vitripennis*.

Species	Whole Genome				PCGs *		tRNAs **		rRNAs ***		Control Region	
	Size (bp)	AT%	AT Skew	GC Skew	Size (bp)	AT%	Size (bp)	AT%	Size (bp)	AT%	Size (bp)	AT%
*A. grahami*	15,621	78.6	0.068	−0.080	10,972	77.5	1426	80.2	1955	82.1	1296	80.4
*A. xanthonota*	15,894	78.4	0.065	−0.082	10,973	77.4	1423	80.0	2015	81.9	1513	79.9
*B. ferruginea*	15,262	76.5	0.170	−0.150	10,974	75.0	1443	79.9	1915	78.4	1006	84.7
*B. qiongana*	15,788	76.9	0.166	−0.136	10,975	75.3	1437	80.0	1929	78.3	1491	84.2
*Ci. viridis*	13,461	78.8	0.059	−0.074	10,976	78.5	1283	79.3	1193	82.1	/	/
*Ci. viridis*	15,880	78.1	0.058	−0.076	10,977	77.0	1425	78.3	1919	80.9	1645	82.4
*Co. yasumatsui*	15,019	77.2	0.089	−0.126	10,978	75.7	1412	79.7	2020	79.7	658	89.5
*Cuerna* sp.	12,597	77.5	0.062	−0.089	10,979	77.1	1352	79.5	313	82.1	/	/
*H. vitripennis*	15,304	78.4	0.097	−0.118	10,980	77.2	1416	78.7	1929	79.7	1033	88.1

* PCGs: protein-coding genes; ** tRNAs: transfer RNA genes; *** rRNAs: ribosomal RNA genes.

## Data Availability

Date available on request.

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
