# Peer review of "Characterization of Two Complete Mitochondrial Genomes of Atkinsoniella (Hemiptera: Cicadellidae: Cicadellinae) and the Phylogenetic Implications"

_insects, 2021, doi:10.3390/insects12040338_

Round 1
Reviewer 1 Report
Dear Editor,
i have carefully read this submission.
The authors present the mitogenomes of two Atkinsoniella species, belonging to a Cicadellidae genus with no mitogenomes available so far. This study is clearly presented, in good English language. Moreover, the analysis seems to have been properly designed and executed. Previous work in the field has been properly acknowledged and phylogenetic includes the appropriate previously published mitogenomes.
I have no major concerns, just see the two below comments that the authors could try to address:
- General introduction comments: the authors point to the importance of mitogenomes in phylogenetics. This is fine, but it would be maybe worth mentioning why it is not ideal to rely only on mitochondrial markers and a combined approach of mitochondrial and nuclear markers (integrative approach) is much more robust. See for example that mitogenome is still a single molecule and there are phenomena leading to selective sweeps of mitochondria as well.
- It is common to detect mitochondrial SNPs even in single individuals when NGS is applied and they could reflect either 'real' heteroplasmy or sequencing/assembling artifacts. How does the authors' pipeline deal with such phenomena?
Reviewer 2 Report
Dr Jiang and Collaborators have obtained the complete sequence of the mitogenome of two species of Cicadellinae. They describe the two new genomes, provide some indications on other genomes present in GenBank but not described thus far, and present a phylogenetic analysis of Cicadellidae.
Generally speaking, the description of new mitogenomes is generally published in specialized journals (with a different impact factor from Insects) unless the new sequences are useful to address some relevant question in terms of genome evolution, phylogeny, etc...
I have the feeling that the main content of this manuscript is a simple genome description. A phylogenetic analysis is indeed presented, but it does not extend significantly over the results of other authors as to identify this as a significant step forward, worth of publication in Insects.
The Editor will give his opinion on the appropriateness of thsi submission in terms of subject matter.
Limited to the study clarity and correctness, I am raising a general point and a number of point issues below.
Analyses/descriptions are presented, with a different level of detail, for three group of taxa: a) the two new mitogenomes; b) 7 mitogenomes present in GenBank but unpublished; c) ~60 additional cicadellids used in the phylogenetic analysis.
More specifically, many comparisons are conducted on the b) group: base composition, repeat structure, start/stop codons, RSCU, etc... Nevertheless, this group of 9 sequences is not a natural assemblage, and it merely stems from the coincidence that some genomes were submitted but not published by other authors. As such, not the most appropriate level to conduct evolutionary comparisons.
My suggestion is to define a more interesting and natural group to conduct genome comparisons, such as Cicadellinae or the entire Cicadellidae. All measurements/statistics/analyses may be conducted for all genomes in the selected group and presented as ranges. These will in turn be the background information against which the two new genomes can be compared/discussed. I would take this as a relevent point in the revision.
Suppl. Table S2: aminoacid models (e.g. BLOSUM) are used for the last dataset (PCGs+RNA) that nevertheless is endoded as DNA. Please clarify and correct.
Lines 46-48: accurate identification is obviously imperative for pest control, while how a phylogenetic tree may help with species identification may be better argumented.
Table 1: the correspondence between subfamily level (column 1) and species (column 2) cannot be read with confidence. I suggest adding horizontal lines to separate subfamilies or to place the subfamily label in line with the first species of each subfamily. Futhermore 'Treehopper' is not a taxonomic identifier at the subfamily leval, please elaborate.
Lines 90-91: please provide geographical coordinates.
Line 100: please specify if the two DNAs were barcoded before sequencing or not. If not, indicate which cautionary steps were taken to avoid cross-assembly of the two genomes, arguably very similar at the sequence level.
Line 104: One single COX1 sequence was used to initiate assembly in the two species. While this is probably technically faesible, how were the two newly assembled sequences assigned to the two species?
Line 116: right boundary of srRNA abuts to CR, how was the exact boundary defined?
Line 127: genomes analysed are a subset of the over 100 genomes available in GenBank for Cicadellidae. Please explicitate the rationale used to select species for the phylogenetic analysis.
Line 147: indicate the starting partitions used to initiate the PartitionFinder analysis.
Line 153: it is not clear if the run was stopped as split frequencies dropped below 0.001 or the full 100 million generations were processed. In the latter case, please explain why 25% burnin started from this point and not from generation 1.
Line 158: it is not clear if annotations for the additional 7 mitogenome sequences (not produced in this study) were revised or taken from GenBank.
Figure 2: please enlarge text within figure and color key.
Table 2: difficult to read, please separate the two genomes. I suggest dividing them vertically.
Line 184: there is an extra space.
Line 201: clarify that 63/64 are sums of multiple overlaps.
Line 214: 3643 aminoacids do not correspond to 10996 bases. Please correct and double check other similar instances.
Figure 4: hollow dots are difficult to see. Furthermore, there are some inconsistencies, such as in Valine where a AU bond in the acceptor arm is not indicated as expected.
Lines 309-327: please try to present the results of the phyogenetic analysis more clearly. If I understand correctly, subfamilies are consistently recovered as monophyletic in different analyses, but relationships between subfamilies are different across analyses. If correct, I suggest rephrasing the entire paragraph with this in mind.
Lines 331-333: please add as figure in the main text similar to figure 7
Figure 7: enlarge text, labels and support values are difficult to read.
Lines 362-363: in the light of the inconsistencies between different analyses, the conclusion section appears very optimistic with regards to the possibility to resolve phylogenetic relationships among Cicadellidae.
Round 2
Reviewer 2 Report
I think that, following the revision made by the Authors on their initial submission, the manuscript is now acceptable for publication in Insects.